

# Impact of the temperature-cloud phase relationship on the simulated Arctic warming during the last interglacial

Nozomi Arima★[1], Masakazu Yoshimori★[1], Ayako Abe-Ouchi[1], Ryouta O'ishi[1], Wing-Le Chan[1], Sam Sherriff-Tadano[2], and Tomoo Ogura[3]

[1]Atmosphere and Ocean Research Institute, The University of Tokyo, Kashiwa, Japan
[2]University of Ryukyus, Okinawa, Japan
[3]National Institute for Environmental Studies, Tsukuba, Japan
★These authors contributed equally to this work.

*Correspondence to*: Masakazu Yoshimori (masakazu@aori.u-tokyo.ac.jp)

**Abstract.** The Arctic during the last interglacial period (LIG) was considered warmer than today. While a recent proxy-based study suggests the disappearance of summer sea ice in the Arctic at the LIG, many climate models fail to capture this feature. It is thus essential to investigate sources of uncertainty in numerical models. The current study examines the impact of the temperature-cloud phase relationship. Sensitivity studies are conducted for the first time to explore the potential importance of this relationship in simulating the LIG climate. Two different cloud parameter sets are used for an atmosphere-ocean general circulation model with and without the dynamic vegetation feedback. The model with cloud parametrization permitting liquid water at a lower temperature and a larger fraction of supercooled liquid water at the same temperature simulates a warmer preindustrial (PI) climate, larger annual mean Arctic warming at the LIG, and substantially reduced sea ice cover during summer at the LIG. It is demonstrated that the low-level clouds play a crucial role in controlling the Arctic response via the greenhouse effect. The result indicates the importance of the temperature-cloud phase relationship in simulating the Arctic climate at the LIG. It also highlights the importance of accurately simulating modern sea ice thickness and representing the processes that affect the fraction of supercooled liquid water in clouds.

## 1 Introduction

The Last Interglacial (LIG) refers to the period from 129,000 to 116,000 years ago, when the Earth was relatively warm and ice sheets over the North American and Eurasian continents were relatively small during the Pleistocene glacial cycles (Gulev et al., 2021). The global mean temperature is estimated to be about 0.5 to 1.5℃ higher than the Preindustrial (PI) (Gulev et al., 2021). The primary source of external forcing at LIG is the Earth's orbital configuration, in which the Northern Hemisphere (NH) summer solstice is closer to the perihelion and the Earth's axial tilt is larger by about 0.6º than PI (Otto-Bliesner et al., 2017). However, it is not fully understood how the seasonal and latitudinal redistribution of insolation causes a warmer climate than today. Indeed, on average, multi-model climate simulations for 127 kaBP yield a negligible difference (-0.02℃) in global annual mean temperature from the PI (Otto-Bliesner et al., 2021).



The Arctic summer at LIG was estimated to be warmer than PI by as much as 4-5℃ (Bennike et al., 2001), and has drawn considerable attention, given that the Arctic is currently warming much faster than the rest of the world, and sea ice is decreasing dramatically. Kageyama et al. (2021) found a positive correlation in summer sea ice area across climate models between equilibrium LIG and transient $CO_2$ increasing experiments, suggesting that the Arctic warming at LIG is potentially

a practical constraint for future projections. While climate models generally simulate summer land warming reasonably well, they tend to underestimate the annual mean Arctic warming compared to what proxy records suggest (Otto-Bliesner et al., 2021). In addition, there is a considerable model spread in the Arctic temperature anomaly from the PI (Otto-Bliesner et al., 2021).

The first comprehensive model-data comparison for the Arctic sea-ice cover was conducted by Kageyama et al. (2021).

Recently, Vermassen et al. (2023) examined the presence or absence of sea ice, including in the central Arctic, based on microfossil assemblages and argued that the Arctic was essentially ice-free in summer at LIG. However, only one of the 12 models analyzed by Kageyama et al. (2021) simulated the ice-free summer Arctic. O'ishi et al. (2021) reported that the dynamic vegetation feedback, in which the vegetation type changes according to the local climate change, helps simulate the substantially warmer Arctic at LIG, although not to the extent of removing the summer Arctic sea ice entirely. Diamond et al.

(2021) reported that the explicit representation of melt pond and consequent decrease in surface albedo in the sea ice model component plays an essential role in 'successfully' simulating the ice-free Arctic summer. Although it may not be fundamental, we note that the model has a relatively high climate sensitivity, and the other model with the explicit melt pond representation does not capture the ice-free condition.

Another potential source of uncertainty in the LIG simulation, which we focus on in this study, is the temperature-cloud

phase relationship in models. Here, the cloud phase refers to cloud particles, which can be either cloud droplets (liquid water) or ice crystals (solid water). It is well known that mixed-phase clouds, composed of both liquid and solid cloud particles, are common in the Arctic (Kay et al., 2016; Morrison et al., 2011), and the cloud phase is one of the essential key parameters influencing climate change (Tan et al., 2016; Tsushima et al., 2006; Yoshimori et al., 2009; Zelinka et al., 2020).

There are two primary effects of phase changes of cloud particles on cloud properties: albedo and lifetime. A warming

increases the fraction of supercooled liquid water (SLF) at the expense of ice crystals, provided that the total amount of water is constant. As the size of cloud droplets is generally smaller than the ice crystals, the SLF increase induces surface cooling through the increase in cloud albedo (Murray et al., 2021). It takes more time for cloud droplets to grow and precipitate, due to their smaller size, than for ice crystals, resulting in a slower auto-conversion rate. In addition, the saturation vapor pressure is higher against the liquid water surface than the ice surface, and ice crystals tend to grow faster at

the expense of cloud droplets (the so-called Wegener–Bergeron–Findeisen process). Therefore, the SLF increase leads to a longer cloud residence time and an increase in cloud amount over a specific time interval. The more clouds there are, the less shortwave (SW) radiation reaches the surface (cooling), while more longwave (LW) radiation is emitted back to the surface (warming). One important conclusion from this reasoning is that the warming effect should dominate in the Arctic winter when insolation is very limited (Tan and Storelvmo, 2019) and Arctic amplification reaches its peak. Therefore, the



representation of cloud phase changes may be important in simulating Arctic climate change at LIG. Furthermore, it is essential to note that the temperature-cloud phase relationship is extremely diverse among models (McCoy et al., 2015). Although it is the first time that the impact of cloud phase representation is investigated for the LIG simulation, the effect has been examined in the context of climate sensitivity (Tan et al., 2016), glacial inception (Sagoo et al., 2021), and Atlantic overturning circulation at the last glacial maximum (Sherriff-Tadano et al., 2023) in previous studies. A broader review on

this topic, as well as Arctic amplification, is covered by Yoshimori et al. (2025) and references therein.

      The primary purpose of this study is to evaluate the effect of cloud-phase temperature dependency on LIG Arctic simulations. This study assumes, for simplicity, that the number of ice-nucleating particles remains constant. It examines the sensitivity of simulated LIG climate to the specified temperature-phase relation. Additionally, we would like to discuss the magnitude of this effect against a range of model spreads. By doing so, we aim to determine whether this is a factor to be

concerned about for the LIG climate simulation.

## 2 Models and cloud parameters

### 2.1 Models

This study employs two climate models: an atmosphere-ocean general circulation model (AOGCM) with and without a dynamic vegetation component, referred to as MIROC4m-LPJ and MIROC4m, respectively. The dynamic vegetation here

means that the vegetation type is computed in the model according to the climate, incorporating the interaction between climate and the geographic distribution of vegetation. We begin with a brief description of MIROC4m. MIROC4m is an AOGCM that consists of the atmosphere, ocean-sea ice, and land surface components interacting with each other through exchanges of energy, water, and momentum (K-1 model developers, 2004). The atmospheric model component is based on the primitive equations, and its horizontal resolution is limited by the T42 spectral truncation (~2.8°×2.8°), with 20 vertical

levels. The land surface component shares the exact horizontal resolution as the atmospheric component, having one canopy layer, five soil layers, and a maximum of 3 snow layers. The ocean model component is based on the primitive equations under the Boussinesq approximation. It has a horizontal resolution of 1.4° in longitude and varies from 0.56° to 1.4° in latitude (with higher resolution toward the equator), comprising 43 vertical levels. The sea ice model component, computing thermodynamic and dynamic processes, shares the exact horizontal resolution as the oceanic component. Fourier filtering is

applied to the ocean grids at NH high latitudes to reduce computational cost due to the convergence of zonal grid spacing toward the North Pole. We must note that this filter smoothes out the large-scale zonal feature of sea ice variables. This climate model runs computationally very efficiently and has been used in many previous studies (e.g., Chan and Abe-Ouchi, 2020; Kuniyoshi et al., 2022; Sherriff-Tadano et al., 2023).

      MIROC4m-LPJ is a model that couples MIROC4m with the dynamic vegetation component (LPJ-DGVM) (O'ishi et al.,

2009; Sitch et al., 2003). A dominant vegetation is represented by the plant functional type and is determined yearly for each grid based on temperature, precipitation, and sunlight averaged over the most recent 20 years. The prescribed atmospheric




$CO_2$ concentration level also affects the plants through the fertilization effect. These variables are passed from the atmospheric component to the LPJ-DGVM, and the diagnosed vegetation type is then passed to the land surface component. By doing so, it is possible to simulate the interaction between climate and vegetation changes. This vegetation-coupled climate model has also been used in previous studies (e.g., Hirose et al., 2025; O'ishi et al., 2021).

## 2.2 Two cloud parameter sets

We examine the difference in climate response to external forcing using two different cloud parameter sets, A and C, which were used by Sherriff-Tadano et al. (2023). We note that the parameter set "B" referred to the parameter set A applied to a different model version in their study as well as in Sherriff-Tadano and Abe-Ouchi (2020), and we retain the names "A" and "C" in this study. Three parameter values are different between the two sets: a) the lowest temperature ($T_{ice}$ in Eq. 1) at which supercooled liquid water droplets can exist; b) a coefficient of autoconversion rate for rain ($\alpha$ in Eq. 2); and c) a coefficient of ice sedimentation rate ($V_0$ in Eq. 3). Note that the actual values are summarized in Table 1.

**Table 1: Perturbed cloud parameter values**

| Cloud parameter set | $T_{ice}$ (°C) | $\alpha$ | $V_0$ |
|---|---|---|---|
| A | -15 | 0.010 | 0.25 |
| C | -28 | 0.025 | 0.30 |

In MIROC4m, SLF, taking the range from 0 to 1, is parameterized in such a way that all cloud particles are solid below $T_{ice}$ and liquid above 0°C, and SLF is linearly interpolated in-between (Ogura et al., 2008):

$$SLF(T) = \begin{cases} 0 & (T \leq T_{ice}) \\ (T - T_{ice})/(0 - T_{ice}) & (T_{ice} < T < 0) \\ 1 & (0 \leq T) \end{cases} \tag{1}$$

where the unit for $T$ and $T_{ice}$ is in °C. The difference between set A and set C in SLF exists for the range between -28 and 0°C, and it is always larger in C, meaning that more cloud liquid water exists in C with a given amount of total cloud water (Fig. 1). The original setting of MIROC4m(-LPJ) follows the temperature dependency of cloud phase in A. Still, C is designed to be closer to recent satellite-based observations (Fig.4 in Sherriff-Tadano et al., 2023).





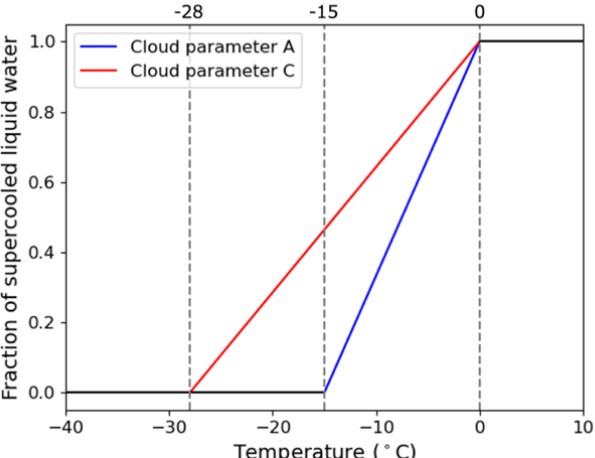

**Figure 1: Fraction of supercooled liquid water in the two cloud parameterizations A and C.**

A complication arises in the coupled-model sensitivity experiments because a single parameter perturbation disturbs the global energy balance of the Earth, leading the model to drift away from the realistic modern climate. To avoid this issue, Sherriff-Tadano et al. (2023) also adjusted the two other cloud parameter values. In MIROC4m(-LPJ), rain rate $P_{rain}$ is

parameterized as

$$P_{rain} = -\frac{\partial l_L}{\partial t} = \frac{\alpha \rho l_L^2}{\beta + \gamma \frac{N_c}{\rho l_L}} + C_c F_p l_L, \tag{2}$$

where $l_L$ is the cloud liquid water content, $\rho$ is the air density, $N_c$ is the cloud droplet number concentration, $F_p$ is the precipitation flux from the layer above, $\alpha, \beta, \gamma$ and $C_c$ are constants (Ogura et al., 2008). Ice sedimentation rate $P_{ice}$ is parameterized as

$$P_{ice} = -\frac{\partial l_F}{\partial t} = \left\{ \frac{V_0 (\rho l_F)^\delta}{\Delta z} \right\} l_F. \tag{3}$$

where $l_F$ is the cloud ice content, $\Delta z$ is the thickness of the model layer, $V_0$ and $\delta$ are constants (Ogura et al., 2008). In the cloud parameter set C, the coefficients associated with autoconversion rate and ice sedimentation rate, $\alpha$ and $V_0$, are both increased compared to set A (Table 1), acting to reduce cloud amount in general.

The snowfall rate is parameterized as

$$P_{ice} = -\frac{\partial l_F}{\partial t} = C_t \left[ 1 - exp \left\{ -\left( \frac{l}{l_c C} \right)^2 \right\} \right] l_F + C_c F_p l_F, \tag{4}$$

where $l$ is the cloud condensate content, $l_c$ the critical cloud condensate content that leads to rapid snow formation, $C$ the cloud amount, and $C_t$ is a constant. We examined the mean lifetime of cloud particles for the autoconversion processes by computing the rate of fractional changes $\frac{1}{l_L} \frac{\partial l_L}{\partial t}$ and $\frac{1}{l_F} \frac{\partial l_F}{\partial t}$ internally in the model for the Arctic lower troposphere and the result is shown in Fig. S1. Both cloud parameter sets show that the cloud liquid water has a longer mean-lifetime than the





cloud ice water. This result is qualitatively consistent with the discussion in the Introduction. Therefore, an increase in SLF is expected to result in a larger cloud amount when averaged over a specific period.

## 3 Experiments

Lists of numerical experiments are presented in Tables 2 and 3 for MIROC4m-LPJ and MIROC4m, respectively. In this manuscript, we primarily present the results of MIROC4m-LPJ, and the results of MIROC4m are added only when

necessary in the Discussion (Sect. 5).

Using the MIROC4m-LPJ, we conducted equilibrium climate simulations for PI and LIG with cloud parameter sets A and C (Table 2). First, we performed the PI experiments with an integration period of 3000 years (PIvA and PIvC). Note that the small letter "v" refers to dynamic vegetation, and "A" and "C" refer to the cloud parameter set. Starting from the quasi-equilibrium states of PIvA and PIvC, we conducted the equilibrium LIG experiments following the PMIP4 protocol (Otto-

Bliesner et al., 2017) with an integration period of 2000 years (LIGvA and LIGvC).

**Table 2: A list of numerical experiments with the MIROC4m-LPJ**

| Name | Boundary conditions (see Table 4) | Cloud parameter set | Integration length (years) |
|---|---|---|---|
| PIvA | PI | A | 3000 |
| PIvC | PI | C | 3000 |
| LIGvA | LIG | A | 2000 |
| LIGvC | LIG | C | 2000 |

We repeated these simulations with MIROC4m (PIfA, PIfC, LIGfA, and LIGfC, as listed in Table 3). Note that the small

letter "f" refers to fixed vegetation. To isolate the effect of dynamic vegetation feedback through a comparison of MIROC4m-LPJ and MIROC4m, the most dominant vegetation type simulated by PIvA over the last 100 years for each grid is prescribed in both PIfA and LIGfA experiments, and that simulated by PIvC is prescribed in both PIfC and LIGfC experiments. This approach does not allow the bias in the simulated modern vegetation to directly influence the comparison between MIROC4m-LPJ and MIROC4m.


**Table 3: A list of numerical experiments with the MIROC4m**

| Name | Boundary conditions (see Table 4) | Cloud parameter set | Integration length (years) |
|---|---|---|---|
| PIfA | PI | A | 2000 |
| PIfC | PI | C | 2000 |
| LIGfA | LIG | A | 2000 |
| LIGfC | LIG | C | 2000 |





Table 4: Boundary conditions (LIG follows the PMIP4 protocol described by Otto-Bliesner et al. (2017))

| Boundary conditions | Eccentricity | Obliquity (°) | Longitude of perihelion from the autumnal equinox (°) | $CO_2$ (ppm) | $CH_4$ (ppb) | $N_2O$ (ppb) |
|---|---|---|---|---|---|---|
| PI | 0.016720 | 23.45 | 102.04 | 285.431 | 863.303 | 270.266 |
| LIG | 0.039378 | 24.05 | 275.41 | 275.000 | 685.000 | 255.000 |

This paper expresses the difference between LIG and PI (LIG - PI) as ΔLIG for brevity. For example, the difference between LIG and PI in experiments with the cloud parameter set A in MIROC4m-LPJ (LIGvA – PIvA) is expressed as ΔLIGvA, whereas that with the cloud parameter set C is described as ΔLIGvC. The last 100 years of each experiment are used in the following analysis for the coupled climate models of MIROC4m-LPJ and MIROC4m.

Additionally, we conducted experiments using the atmospheric component of MIROC4m, an atmospheric GCM, to isolate
the impact of three individual parameters in the cloud parameter sets separately. The lower boundary conditions, i.e., SST and sea ice, are taken from the coupled models. The details of the experimental design and its results are described in Appendix A.

## 4 Analysis method

### 4.1 Calendar adjustments

Since the shape and precession phase of the Earth's orbit around the Sun are different at LIG from PI, the angular velocity, which depends on the Sun-Earth distance, differs between the two experiments for each season. The astronomical season may be defined by the angle from the moving vernal equinox, e.g., 90 degrees for the summer solstice. Applying a modern calendar for both LIG and PI results in a comparison of slightly shifted seasons. To minimize this undesired effect, we used the calendar adjustment introduced by Bartlein and Shafer (2019) for the monthly mean values. This adjustment enables us
to more accurately evaluate the seasonal difference between LIG and PI, which is essential for comparing the simulation with proxies whose records are dominated by a specific season.

### 4.2 Partial surface temperature change

We applied the surface energy balance analysis to quantify the contribution of individual feedback processes to the surface temperature (ST) change, following Lu and Cai (2009). This method converts each energy flux term in W m$^{-2}$ between the
two experiments to partial surface temperature change in K, whose total sum amounts to the simulated surface temperature change under an excellent approximation.

The energy balance equation at the surface may be given by

$$Q = (1 - \alpha)S^{\downarrow} + F^{\downarrow} - F^{\uparrow} - LE - H \tag{4}$$



where $S^\downarrow$ is downward SW radiation, and $F^\downarrow$ and $F^\uparrow$ are downward and upward LW radiation, respectively. $H$ and $LE$ are

sensible and latent heat fluxes (positive upward), respectively, and $Q$ is the heat storage rate in the subsurface (e.g., heat conduction into the soil layers).

The perturbation equation for the difference between the two experiments, denoted by $\Delta$, is written as

$$4\sigma\bar{T}^3\Delta T \approx \Delta F^\uparrow = -\Delta\alpha\bar{S}^\downarrow - \Delta\alpha\Delta S^\downarrow + (1 - \bar{\alpha})\Delta S^{\downarrow,clr} + \Delta F^{\downarrow,clr}$$

$$+(1 - \bar{\alpha})\Delta S^{\downarrow,cld} + \Delta F^{\downarrow,cld} - \Delta LE - \Delta H - \Delta Q \tag{5}$$

The overlines denote the average of the two paired experiments for a comparison. Here, SW and LW radiation are decomposed into clear-sky and cloud (= total sky – clear sky) radiative effects. Dividing by $4\sigma\bar{T}^3$ on both sides, the surface temperature change is expressed as the sum of 9 partial-change terms on the right side. It is symbolically written as

$$\Delta T = alb + alb * SW + SW_{clr} + LW_{clr} + SW_{cld} + LW_{cld} + evap + sens + subsurf \tag{6}$$

The physical meaning of each term is summarized in Table 5. While this diagnosis is made explicitly for the ST change,

rather than the surface air temperature (SAT) change, it is also helpful to understand the latter because these two variables are thermally coupled. Indeed, it is confirmed that changes in these two variables are nearly the same in all pairs of experiments compared in this study. Nevertheless, care must be exercised in the physical interpretation. For example, upward sensible heat flux cools the surface yet warms the air above. The upward heat flux from the subsurface to the surface represents the warming effect on the surface, which may be transferred to sensible and latent heat fluxes, warming the air

above. Therefore, a warming from the subsurface term and a cooling through evaporation and sensible terms imply atmospheric warming through the release of heat from the ocean.

**Table 5: Physical meaning of individual terms in the analysis of partial surface temperature change**

| Symbols | The effect of perturbed component at the surface |
|---|---|
| $alb$ | Albedo |
| $alb * SW$ | Synergy of albedo and downward SW radiation |
| $SW_{clr}$ | Clear-sky downward SW radiation |
| $LW_{clr}$ | Clear-sky downward LW radiation |
| $SW_{cld}$ | SW cloud radiative effect |
| $LW_{cld}$ | LW cloud radiative effect |
| $evap$ | Latent heat due to evaporation |
| $sens$ | Sensible heat |
| $subsurf$ | Heat storage in the subsurface |




# 5 Result

## 5.1 Impact of cloud parameterization on the preindustrial climate simulation

Figure 2 shows the difference in annual mean SAT between PIvC and PIvA, highlighting the impact of cloud parameterization on preindustrial simulations. The global, annual mean SAT is 13.9°C and 14.6°C in PIvA and PIvC, respectively. Note that crude comparisons with a global atmospheric reanalysis dataset are presented in Fig. S2. PIvC is generally warmer than PIvA except for the regions around Antarctica. This spatial pattern difference is similar to what was

shown in Sherriff-Tadano et al. (2023), in which observed modern vegetation is prescribed. The warmer tropics in PIvC than PIvA are attributable to more solar radiation reaching the surface due to less cloud cover (not shown), which is caused by the higher efficiency of autoconversion rate with the cloud parameter set C (Sect. 2.2). In the Arctic region, defined here as north of 60°N, PIvC is warmer than PIvA by 1.1°C. There is, however, little difference in the simulated vegetation distribution between PIvA and PIvC (Figs. 3a and 3b). We note that the PIvA vegetation is consistent with O'ishi et al. (2021), which

used the identical cloud parameterization but a slightly different value for the ocean's eddy isopycnal thickness diffusivity of the Gent-McWilliams parameterization. Factors for the Arctic SAT difference between the two cloud parameterizations were not investigated in Sherriff-Tadano et al. (2023), where the focus was on the Southern Ocean. Figure 4 shows the partial contribution of individual components to the total ST difference in the Arctic, as diagnosed by the diagnosis method described in Sect. 4.2. Note that the SAT difference is almost indistinguishable from the ST difference (not shown). The ST

difference stands out from October to January, with a peak in November (2.8°C). On the contrary, the difference is diminished during summer. The figure shows a dominant positive contribution from the LW cloud radiative effect (CRE), although the SW CRE is not negligible in some months.

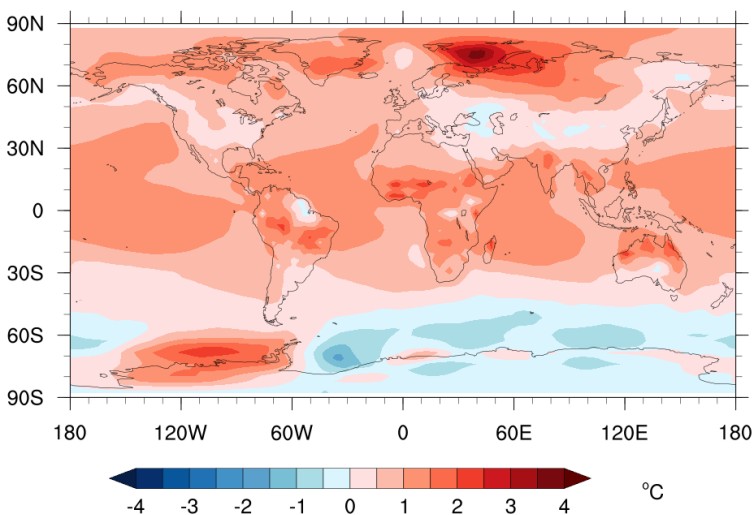

**Figure 2: Difference in annual mean surface air temperature between PIvC and PIvA (°C).**



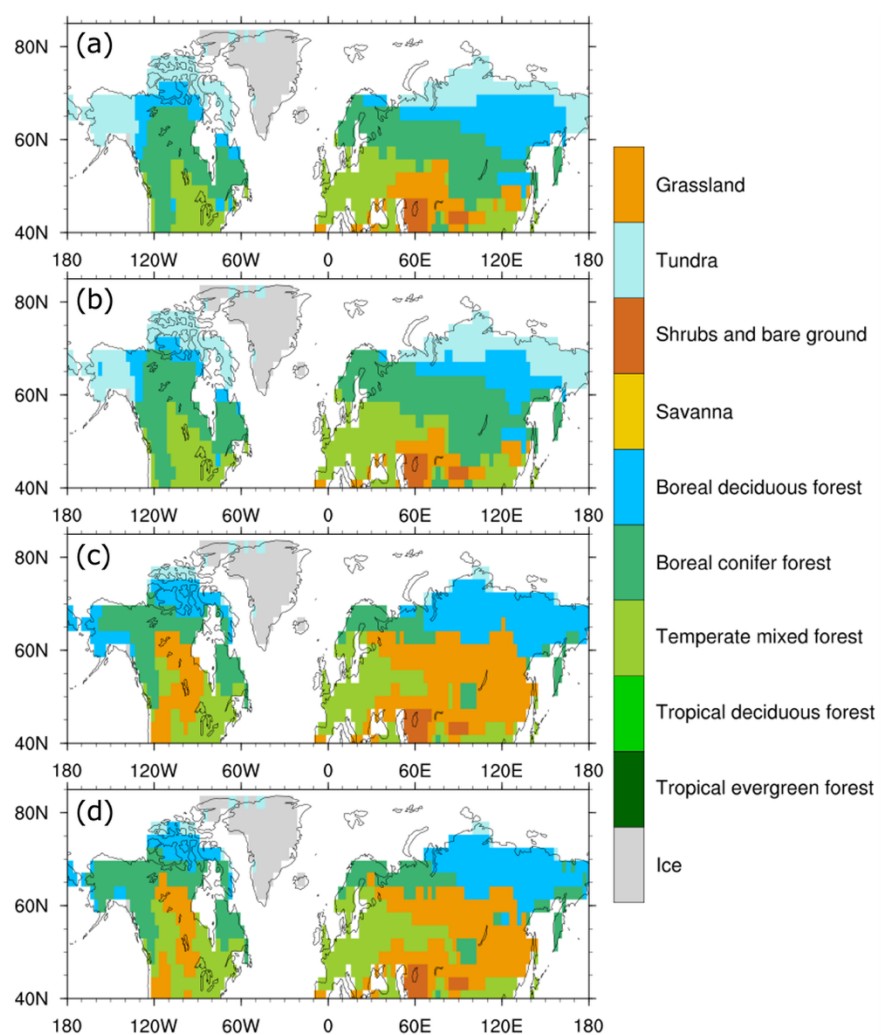

**Figure 3: Simulated vegetation distribution: (a) PIvA; (b) PIvC; (c) LIGvA; and (d) LIGvC.**






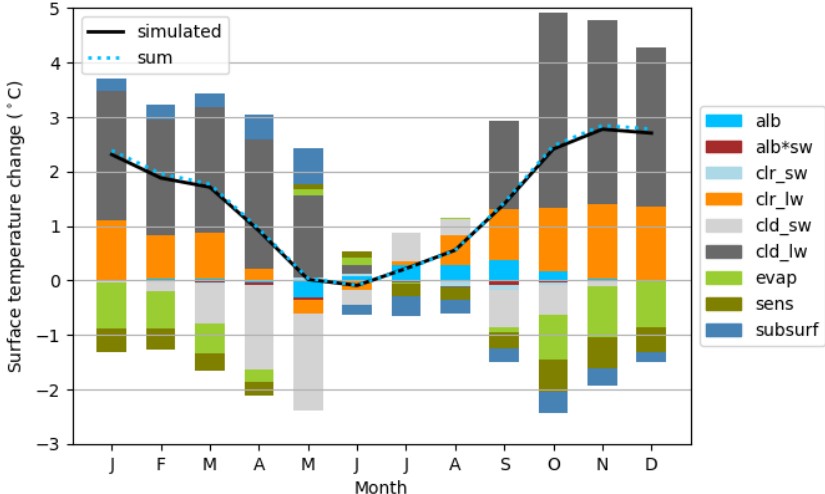

**Figure 4: Attribution of Arctic (60-90°N) ST difference to feedback components between PIvC and PIvA (°C). Please refer to Table 5 for a description of each component. The solid black polygonal line denotes simulation, and the dashed blue line indicates the sum of the diagnosed partial temperature differences.**

Figure 5 shows annual mean differences in cloud liquid water path (LWP) and low-level cloud amount for the Arctic between PIvC and PIvA. Except for the northern North Atlantic, larger LWP and low-level cloud amount are simulated in PIvC than PIvA. The larger LWP in PIvC was expected as SLF is parameterized to be larger below 0°C in PIvC compared to PIvA (Sect. 2.2). As the precipitation efficiency is significantly lower for cloud water than for cloud ice in the model (Fig. S1), the average lifetime of cloud water becomes longer. Consequently, the Arctic low-level cloud amount also increases in

PIvC compared to PIvA. The final simulated difference, of course, is modified by feedback, such as changes in sea ice cover. The difference in low-level cloud amount is far more pronounced than that of middle and high clouds (not shown). Additionally, the downward LW radiation from low-level clouds is more effective in warming the surface, as the emission temperature at lower altitudes is higher than at higher altitudes. The dominant influence of temperature-cloud phase parameter, rather than other perturbed cloud parameters was verified with additional AGCM experiments and described in

Appendix A.




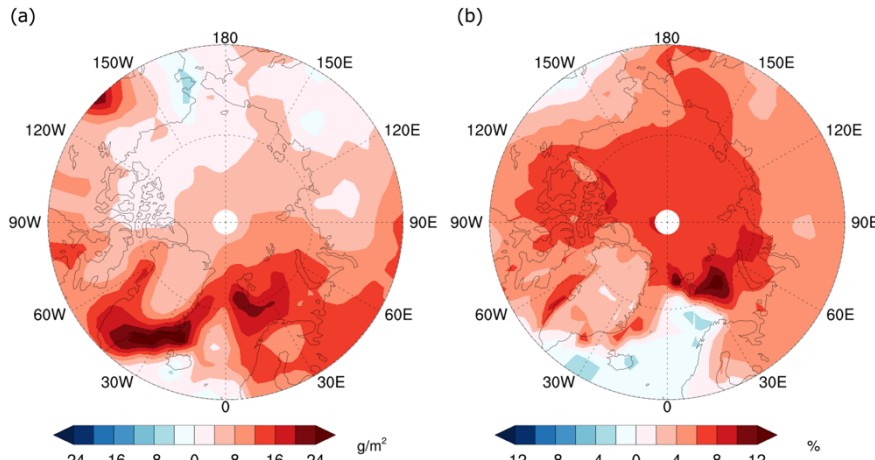

**Figure 5: (a) Difference in annual mean liquid water path between PIvC and PIvA (g/m²); (b) same as in (a) but for low-level cloud amount (%).**

Figures 6a and 6b show September sea-ice concentrations for PIvA and PIvC, respectively. The smaller area of sea ice cover in PIvC than PIvA is consistent with the warmer Arctic in PIvC. Additionally, the sea ice of PIvC is overall thinner than that of PIvA, with a difference of up to 30 cm in the Central Arctic (Figs. 7a and 7b). This difference in sea ice thickness is later shown to be critically important to understanding the sea ice distribution difference in LIG simulations.

    The cold bias and excessive sea ice cover in the Barents and Kara Seas show some improvement in PIvC compared to
PIvA, based on sea ice and global atmospheric reanalysis datasets (HadISST2 and ERA5 in Fig. S2, respectively). We note, however, that both versions suffer equally from a significant warm bias over North America (Fig. S2) and a lack of Arctic sea ice along the North American coast, regardless of the applied cloud parameterization.



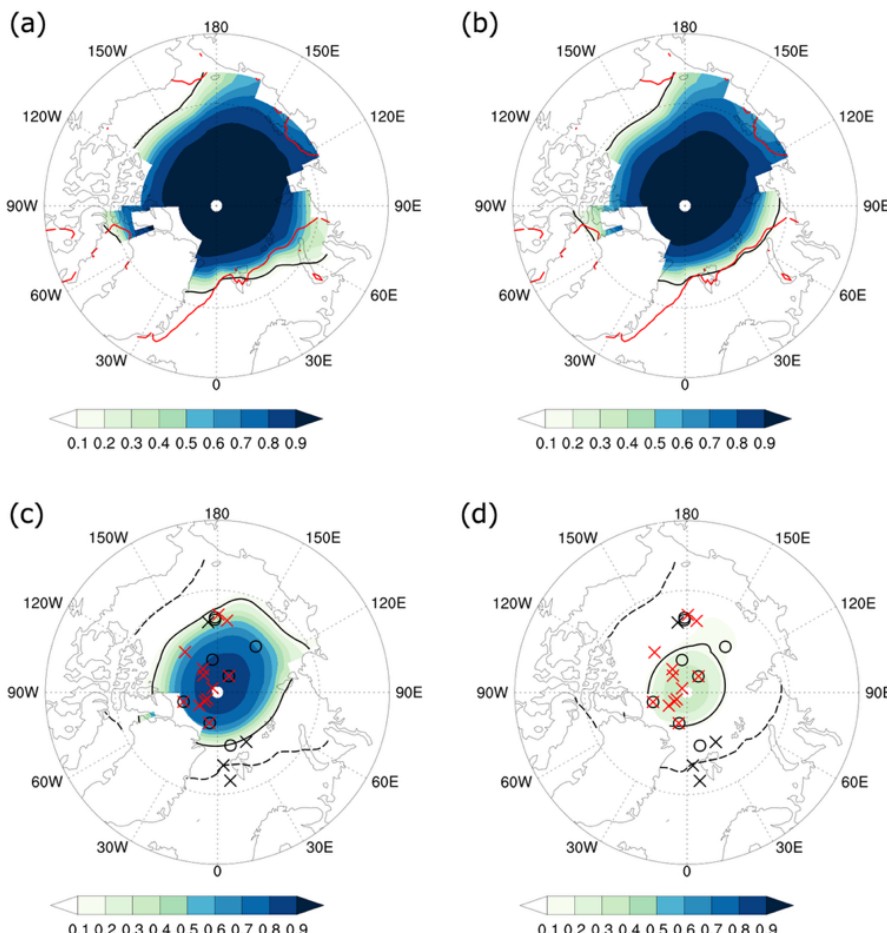

**Figure 6: September sea-ice concentration in the Arctic: (a) PIvA; (b) PIvC; (c) LIGvA; and (d) LIGvC. In (a) and (b), black lines denote the simulated boundaries for the ice concentration of 0.15, while red lines denote the observed ones (HadISST2). In (c) and (d), black solid lines denote the simulated boundaries for the ice concentration of 0.15, while black dashed lines denote those in the corresponding preindustrial simulations. The year-round ice cover (circles) and summer ice-free conditions (crosses) suggested by proxies are also plotted: Black symbols for Kageyama et al. (2021) and red symbols for Vermassen et al. (2023).**



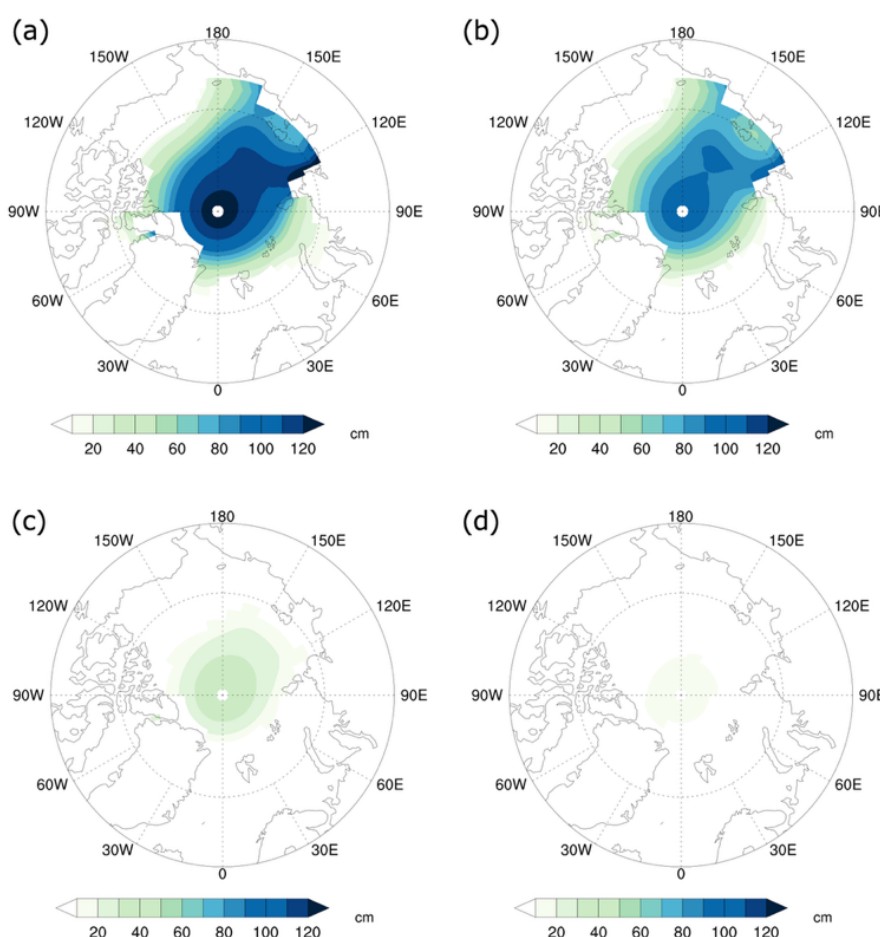

Figure 7: September sea-ice thickness (cm) in the Arctic: (a) PIvA; (b) PIvC; (c) LIGvA; and (d) LIGvC.

## 5.2 Comparison of LIG simulations with proxies

Figures 8a and 8b show the difference in annual mean SAT for ΔLIGvA (LIGvA – PIvA) and ΔLIGvC (LIGvC – PIvC),
respectively. In both cases, simulated LIG is warmer than PI in the Arctic (to the north of 60°N) by 2.7°C for ΔLIGvA and
by 3.1°C for ΔLIGvC. The simulated temperature increase from PI to LIG is slightly larger with the cloud parameter set C,
which is closer to the values indicated by the proxies in some locations. However, proxies suggest a warming of around 5°C
along the Arctic coast of Alaska, and more than 8°C in eastern Siberia, whereas only 2-4°C warming is simulated for
ΔLIGvC (1-3°C warming for ΔLIGvA). Thus, a significant gap remains between the simulations and the proxy-based
estimate of annual mean warming at LIG compared to PI. It is important to note that the LIG temperature reconstruction by
Turney and Jones (2010) is compiled from the locally warmest time within a wide period of approximately 13,000 years,
from 129 to 116 kaBP. Thus, it tends to overestimate the 127 kaBP warming rather than underestimate it.



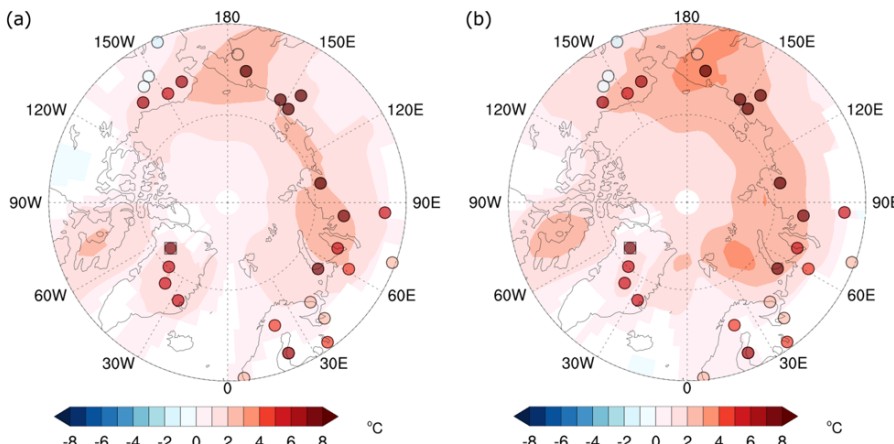

**Figure 8: Comparison of simulation with proxies for the ΔLIG annual mean SAT difference: (a) ΔLIGvA (=LIGvA–PIvA); (b) ΔLIGvC (=LIGvC–PIvC) Only grids where the difference is significant at the 5% level are colored, applying the Student's t-test for 100 samples from the last 100 years of each experiment. The circles are the proxies from Turney and Jones (2010), and the squares are the proxies from Capron et al. (2017).**

Figures 6c and 6d show September sea-ice concentration for LIG simulations, LIGvA and LIGvC, respectively. The area of sea ice cover decreases compared to the corresponding PI simulations. The reduction of sea ice cover in LIGvC is, however, much more drastic than that in LIGvA. Figures. 6c and 6d compare two different sea ice cover reconstruction datasets with these LIG simulations: Kageyama et al. (2021) and Vermassen et al. (2023). At two locations (PS2757: 81.19N, 140.04E and PS92/039-2: 81.92N, 13.83E) identified as ice-covered in summer by the Kageyama et al. (2021) compilation, sea ice is present in LIGvA but absent in LIGvC simulations. On the contrary, overall simulated sea ice cover in LIGvC is much closer to the Vermassen et al. (2023) compilation than LIGvA is, which suggests a nearly ice-free summer Arctic. Notice that the difference in ice thickness in the Central Arctic between LIGvA and LIGvC is as much as 27 cm (Figs. 7c and 7d), which is close to the difference between PIvA and PIvC (Figs. 7a and 7b). This implies that whether sea ice remains in summer at LIG in these simulations is sensitive to the simulated sea ice thickness under modern conditions. In addition, Arctic warming for ΔLIGvC (LIGvC – PIvC) is larger than that for ΔLIGvA (LIGvA – PIvA), which further helps to simulate a smaller summer sea ice cover in LIGvC.

**5.3 Causes of the Arctic warming at LIG**

As discussed in Sect. 5.2, there is a discernible SAT difference between ΔLIGvC and ΔLIGvA. Nevertheless, the simulated vegetation distributions for LIGvA and LIGvC are grossly similar, in which the area covered by tundra at NH high latitudes in PI simulations is replaced by boreal deciduous forest in LIG simulations (Figs. 3c and 3d). Similarly, the area covered by boreal conifer forest at NH mid-latitudes in PI simulations is replaced by grassland in LIG simulations. The former change is particularly effective in altering the surface albedo and consequent warming as discussed by O'ishi et al. (2021).




Figure 9a shows the partial contribution of individual components to the total ST difference in the Arctic for ΔLIGvA according to the diagnosis method described in Sect. 4.2. LIGvA is warmer than PIvA in the Arctic from April to December, with a peak warming in October. From May to August, the dominant warming factors are changes in downward clear-sky SW and LW radiation, as well as surface albedo. The increase in clear-sky SW radiation is expected and reflects the difference in summer insolation at the top of the atmosphere (TOA) due to variations in astronomical parameters. The increase in clear-sky LW radiation is caused by increased air temperature and/or water vapor. We note that the contribution of LW CRE to surface warming is not negligible from September to November, consistent with the increase in low-level cloud amount in autumn (Fig. 10a).

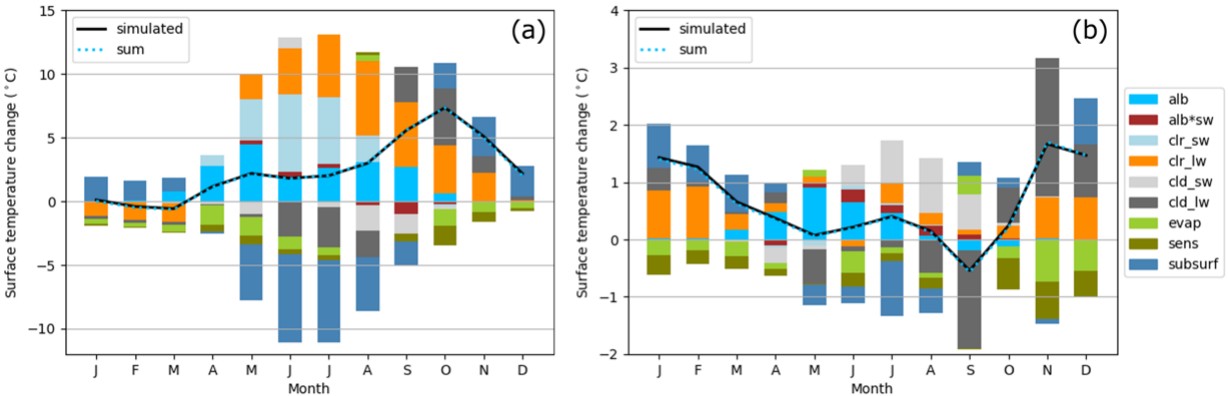

**Figure 9: Attribution of Arctic (60-90°N) surface temperature difference to feedback components (°C): (a) ΔLIGvA (LIGvA–PIvA); and (b) ΔLIGvC – ΔLIGvA. Please see Table 5 for the description of each feedback component. The solid black polygonal line denotes simulation, and the dashed blue line represents the sum of the diagnosed partial temperature differences.**

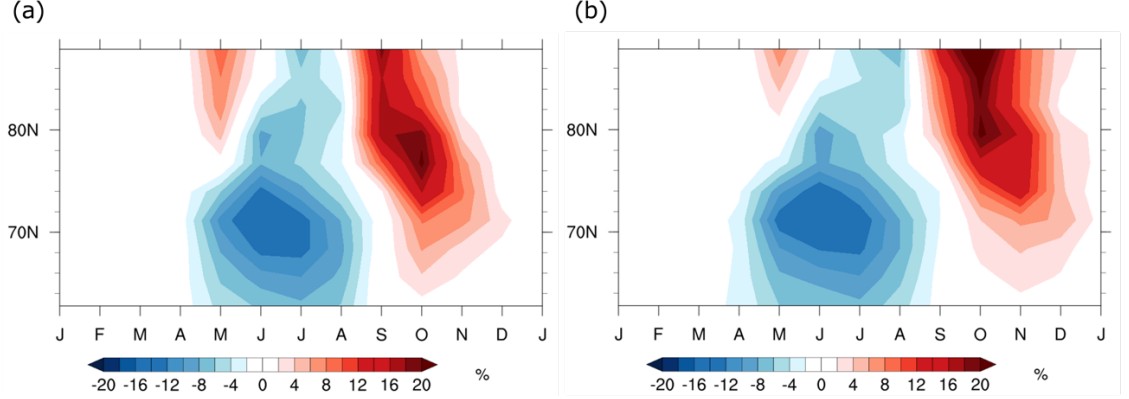

**Figure 10: Difference in monthly mean low-level cloud amount between LIG and PI: (a) ΔLIGvA; and (b) ΔLIGvC.**





Figure 11 shows the changes in sea ice concentration in August and September and low-level cloud amount in September and October for ΔLIGvA (north of 60°N). It appears that regions of sea ice retreat in August and September loosely coincide with the areas of cloud increase in September and October, respectively. The match is firmer in the zonally elongated region to the north of Spitsbergen and the region from the Laptev Sea to the Canadian archipelago in Figs. 11b and 11d. While we do not entirely understand the reason for this one-month delayed response, the previous study by Abe et al. (2016) discussed

a similar relationship between shrinking sea ice and increasing cloud cover from the 1976-2005 simulation using a different version of the MIROC model. In their study, a decrease in sea ice cover in September leads to enhanced upward heat and moisture fluxes through the open water surface. As the air-sea temperature difference increases in October, these fluxes further intensify, resulting in an apparent one-month lag between the decrease in sea ice and the increase in cloud cover. Although radiative forcing and climate conditions differ from ours, Fig. 11 is qualitatively consistent with their argument.


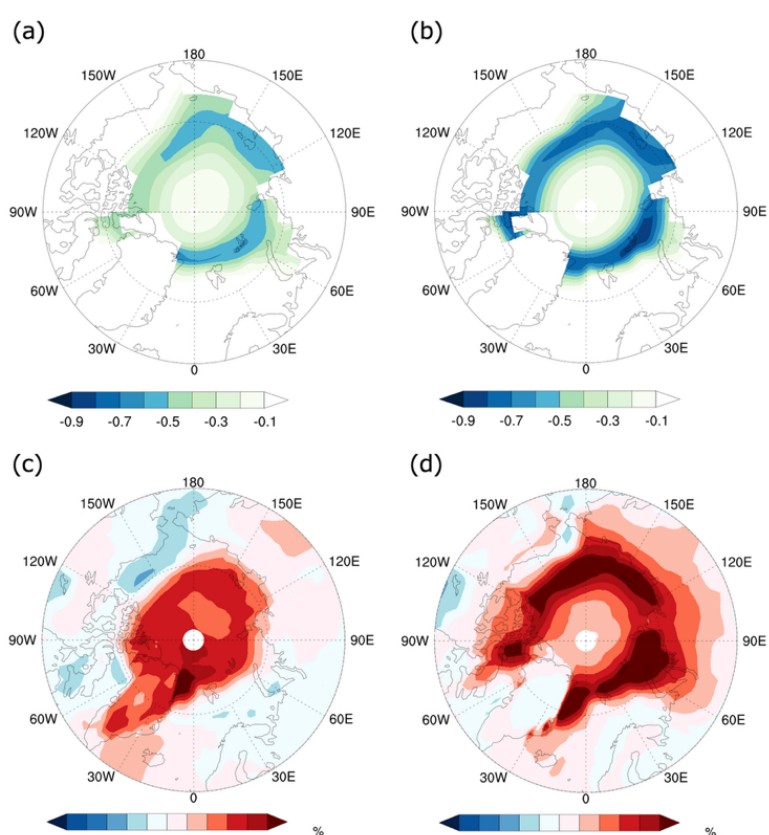

**Figure 11: Changes in sea ice concentration and low-level cloud amount for ΔLIGvA: (a) August ice concentration; (b) September ice concentration; (c) September cloud amount; and (d) October cloud amount.**





### 5.4 Impact of cloud parameterization on LIG climate simulations

Figure 9b shows the partial contribution of individual components to the total ST difference in the Arctic for the difference between ΔLIGvC (=LIGvC – PIvC) and ΔLIGvA (=LIGvA – PIvA). From October to December, the difference between ΔLIGvC and ΔLIGvA is caused primarily via downward LW CRE, consistent with a marked difference in low-level cloud amount increase in ΔLIGvC (Fig. 10b) than ΔLIGvA (Fig. 10a). Figure 12 shows the difference of the difference, i.e., ΔLIGvC-ΔLIGvA, in LWP and low-level cloud amount for the average of October-December. The overall positive

anomalies are observed. As discussed above, the LWP and low-level cloud amount increases in LIG simulations are likely associated with decreased sea ice cover from PI simulations. As the reduction in sea ice cover is much larger in ΔLIGvC than in ΔLIGvA, a larger cloud response occurs in ΔLIGvC than in ΔLIGvA. In addition, the air temperature reaches below -15°C in November and December for large areas of the Arctic, even near the surface at LIG. As the cloud parameterization allows SLF to have non-zero values below -15°C only in the cloud parameter set C, the distinct increase in LWP in the LIG

simulation compared to the PI simulation is observed for ΔLIGvC in November and December. Therefore, the positive anomalies in Fig. 12 are simulated due to a larger decrease in sea ice cover and the allowance of mixed-phase clouds at lower temperatures by the cloud parameter set C. The dominant influence of temperature-cloud phase parameter, rather than other perturbed cloud parameters was verified with additional AGCM experiments and described in Appendix A.

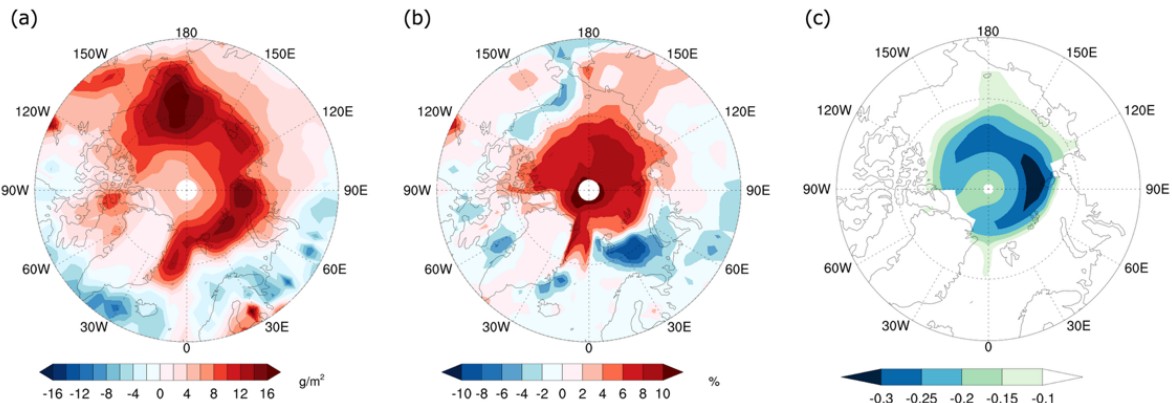


**Figure 12: Difference between ΔLIGvC (LIGvC–PivC) and ΔLIGvA (LIGvA–PivA) from October to December: (a) liquid water path; (b) low-level cloud amount; and (c) sea ice concentration.**

### 6 Discussions

Is the representation of cloud phase also crucial for the model without dynamic vegetation feedback? Many CMIP6-PMIP4

climate models do not account for geographical changes in vegetation types. Here, we examine whether the temperature dependence of cloud phase has a similar impact in such models to what we observed above. We have confirmed that the MIROC4m model with the cloud parameter set C shows a warmer Arctic than the model with the parameter set A for the PI





simulations (PIfC and PIfA, respectively), and thinner sea ice thickness in September by as much as 27 cm in the Central Arctic (Figs. S3), slightly less than the difference between PIvC and PIvA. Regarding the changes from PI to LIG (ΔLIG),

the temperature increase was suppressed to 1.7°C for ΔLIGfA (=LIGfA – PIfA) and 1.8°C for ΔLIGfC (=LIGfC – PIfC), due to the lack of vegetation feedback. Nevertheless, it is common for the winter warming over the Arctic Ocean to be larger in ΔLIGfC due to the increase in cloud amount, and for the sea ice area to decrease significantly in summer at LIG with the cloud parameter set C (Figs. S4 and S5). Therefore, even if many models do not incorporate a dynamic vegetation component, the temperature dependency of cloud phase can be an essential uncertain factor for LIG simulations.

What is the relative importance of cloud phase representation and dynamic vegetation feedback, and how do they affect the model spread at LIG? We added four values of summer (July-September) sea ice area (LIGvA, LIGvC, LIGfA, and LIGfC) to the multi-model data shown in Fig. 13. The summer Arctic sea-ice area in LIGvC is the second smallest, after that of HadGEM3-GC31-LL, the only model that reproduces an ice-free summer Arctic with explicit melt-pond representation. Sea ice areas in LIGvA and LIGfC are relatively close to the LIG sea ice values in CESM2 and NESM3. These results

suggest that the temperature dependency of cloud phase may contribute to part of the model spread for LIG simulations. Additionally, the magnitude of the cloud phase effect is comparable to, or even larger than, the impact of dynamic vegetation feedback. It is suggested that cloud phase and vegetation feedback are essential components to be investigated in addition to the role of melt ponds. It is important to note that summer melting is necessary for melt-pond feedback. In contrast, autumn-winter warming is essential for cloud phase feedback, suggesting different processes are at work in both cases in reducing

sea ice cover at LIG.

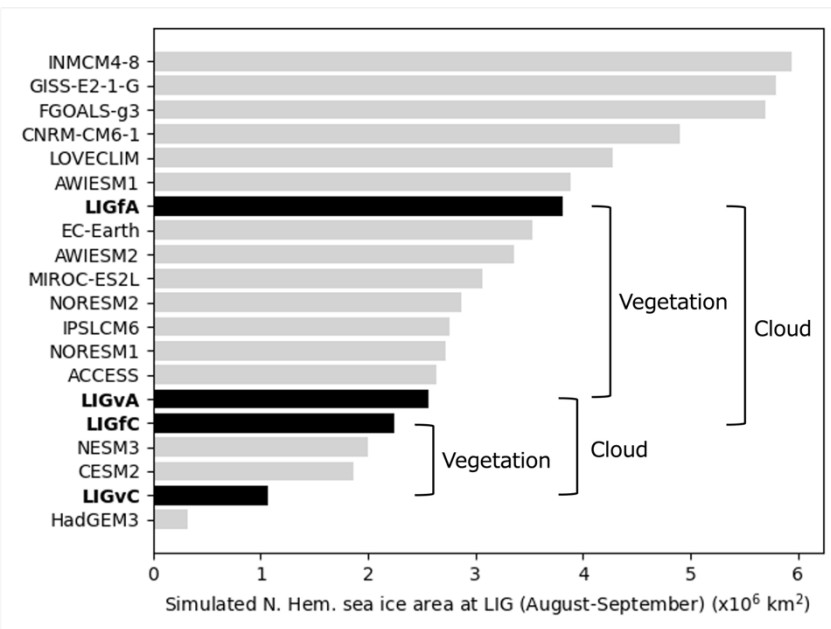



**Figure 13: Simulated NH sea ice area at LIG (August-September). The values of other models (gray bars) are taken from the Supplementary data of Kageyama et al. (2021). "Vegetation" and "cloud" refer to the effects of dynamic vegetation feedback and cloud parameterization, respectively, as examined in the present study.**

One may wonder why the impact of cloud parameterization on the control simulations differs between the Arctic and Antarctic regions. When the cloud parameter set is changed from A to C, the annual mean SAT increases in the Arctic. At the same time, it decreases in many parts of the Antarctic region, except for the 60°W-150°W longitude range (Fig. 2). We investigated this asymmetric response. The feedback analysis described in Sect. 4.2 is also applied to the Antarctic region (60°S-90°S, excluding longitudes 60°W-150°W). It is found that the difference in downward SW CRE during December-February, the summer season in Antarctica, is a dominant contributing factor for the decrease in ST (not shown). In the summer (June to August), Arctic air temperature in most regions with low clouds (below 680 hPa) exceeds 0°C in the two PI experiments. Thus, the difference in temperature dependency of cloud phase between PIvA and PIvC has little effect. On the contrary, in the Antarctic summer (December to February), the air temperature in most regions where low clouds exist is below 0°C, and hence the difference in temperature dependency of the cloud phase has a profound impact on SLF and cloud amount. SLF is larger for PIvC in this temperature range, resulting in more low-level clouds that reflect strong summer sunshine at the high latitudes of the Southern Hemisphere. In short, the asymmetric polar response to the temperature dependence of the cloud phase originates from the different climatological temperatures between the two polar regions.

This study examines the sensitivity of LIG simulations to the temperature-cloud phase relationship, and we do not claim that either of the model versions is more realistic. Nevertheless, the result may be affected by biases in modern climate simulations. It is not easy to judge which model yields more realistic present-day sea ice thickness due to the zonal distribution resulting from the use of polar Fourier filtering. Both versions fail to reproduce generally thicker Arctic ice near the Canadian archipelago and Greenland. While the cloud distribution was examined in Sherriff-Tadano et al. (2023), the absence of a COSP simulator (Bodas-Salcedo et al. 2011) in this model version prevents a rigorous comparison of cloud phases with satellite observations. As stated already, the model suffers from a significant warm bias over North America and, to a lesser degree, in other parts of the world. While the bias is relatively minor over the Arctic Ocean, where the current study focuses, the bias may affect the comparison with terrestrial proxies at LIG. These issues need to be reexamined in the future.

## 7 Summary and Conclusions

For the first time, we investigated the influence of cloud phase representation in LIG simulations by comparing two different temperature-dependent relations for SLF. In the cloud parameter set A, the liquid phase cloud can exist only above -15°C, while it can exist as low as -28°C in the cloud parameter set C. Consequently, the SLF is always larger between -28°C and 0°C, and the cloud phase change occurs with a slight temperature perturbation even below -15°C, with the parameter set C.





The AOGCM with a dynamic vegetation component, MIROC4m-LPJ, exhibits a warmer Arctic climate for parameter set C than for A, primarily because larger LWP and low-level cloud amount lead to a more substantial cloud-induced greenhouse effect in winter. This mechanism is unique to the polar regions, where mixed-phase clouds prevail during cold seasons with limited sunshine. While there is a relatively small difference in sea ice concentration between the two versions of the model, a notable difference exists in sea ice thickness of up to 30 cm.

LIG simulations with MIROC4m-LPJ show slightly larger warming with parameter set C than with parameter set A, compared to the respective PI simulations, resulting in a marginally better agreement with temperature proxies for the model with parameter set C. The larger warming is attributed to the larger LWP and low-level cloud amount due to the cloud phase feedback from October to December. The positive cloud phase feedback is larger with the parameter set C because the reduction in sea ice is larger, and the phase change occurs below -15°C only for the parameter set C. With the thinner sea ice

at the PI simulation and larger warming at LIG from the PI simulation for the parameter set C, sea ice cover at LIG in September is much smaller with the parameter set C. While the extent of sea ice cover at LIG is inconclusive from two different proxy-based compilations, the parameter set C shows a much better agreement with the new compilation, which suggests an ice-free Arctic in summer.

Since many CMIP6-PMIP4 climate models do not incorporate dynamic vegetation feedback, we also examined the impact

of cloud phase representation using the AOGCM without the dynamic vegetation component, MIROC4m. Although the levels of warming and reduction of sea ice extent are much more moderate in MIROC4m compared to MIROC4m-LPJ, a similar impact of cloud phase representation is observed, suggesting that the cloud phase representation might be one of the factors that generate differences in multi-model LIG simulations.

Previous studies have highlighted the importance of representing vegetation change in simulating significant annual-mean

Arctic warming (O'ishi et al., 2021) and melt ponds on sea ice in simulating ice-free summer Arctic at LIG (Diamond et al., 2021). This study points out an additional source of uncertainty that may help alleviate the underestimated Arctic warming by current climate models. As the temperature dependency of the cloud phase is controlled by the availability of ice-nucleating particles (and cloud condensation nuclei) in the real world, no universal cloud-phase function of temperature is expected to exist. The relation is likely to vary with climate state and regions. Therefore, this issue cannot be resolved solely

by present-day observations. We argue that refining cloud phase formulation and improving the reproducibility of present-day sea ice thickness in climate models are essential to improve Arctic simulation at LIG and are likely to be crucial for the future simulations.

**Code and data availability**

The codes for MIROC4m and MIROC4m-LPJ are not publicly archived because of the copyright policy of the MIROC

community. Readers are requested to contact the corresponding author, if they wish to validate the model configurations of

MIROC models and conduct replication experiments. We will make the data publicly available for reproduction of the figures upon acceptance.

**Author contribution**

NA and MY are joint first authors for this paper and prepared the manuscript with contributions from all coauthors. MY designed the experiments, and NA and MY carried them out. NA conducted most of the analysis, and MY finalized the figures. This paper is based on NA's master's thesis, which MY supervised.

**Competing interests**

The authors declare that they have no conflict of interest.

**Acknowledgments**

All computations were performed on the JAMSTEC Earth Simulator 4 supercomputer. We thank the MIROC model development team for making the model available, and Drs. Takashi Obase and Fuyuki Saito for their technical assistance in importing the model to the Earth Simulator 4. The AI tool Grammarly was used to fix grammatical errors in English.

**Financial support**

This work was supported by the Arctic Challenge for Sustainability Programs, ArCSII (Grant No. JPMXD1420318865) and ArCSIII (Grant No. JPMXD1720251001), and a Grant-in-Aid from JSPS KAKENHI (Grant No. JP19H05595 and JP24H00256).

**Appendix A: AGCM experiment**

As stated in the main text, we conducted sensitivity experiments using an AGCM component of the MIROC4m to isolate the effect of a perturbed cloud parameter ($T_{ice}$ in Eq. 1) controlling the temperature-phase relationship from the effect of other perturbed cloud parameters ($\alpha$ and $V_0$ in Eqs. 2 and 3) controlling the autoconversion rate and ice sedimentation rate. The influential parameter was identified by partially swapping these parameters. The list of experiments is presented in Table A1. An AGCM, instead of an AOGCM, was used to prevent the model from drifting away due to the Earth's energy imbalance. In Table A1, "Boundary conditions" refers to sea surface temperature, sea ice, and vegetation. Each AGCM experiment lasts 15 years, and the last 10 years were used for analysis.





We begin by examining the differences between the two cloud-parameter sets in PI simulations. Fig. A1a, representing the differences in low-level cloud amount simulated by the AOGCM, is reproduced reasonably well in Fig. A1b, representing the same field simulated with the AGCM. However, the experiment in which only $T_{ice}$ is exchanged (Fig. A1c) fails to capture the main feature of the difference (cloud amount is reduced significantly instead of increasing), suggesting that the parameter $T_{ice}$ plays a key role. On the other hand, the experiment in which $\alpha$ and $V_0$ are exchanged (Fig. A2d) agrees with

the sign of the complete response (Fig. A1b), suggesting that the result is qualitatively insensitive to these parameters. These results support our interpretation that the temperature-cloud phase relationship is the dominant factor in determining the difference between the two PI simulations of MIROC4m-LPJ.

    Next, we pay attention to the difference between the two cloud-parameter sets in ΔLIG (LIG-PI). Fig. A2a, representing the differences in low-level cloud amount simulated by the AOGCM, is reproduced reasonably well in Fig. A2b,

representing the same field simulated with the AGCM. Like the case of the difference in PI simulations, Fig. A2c fails to capture the main feature of the difference. Fig. A2d, on the other hand, agrees with the sign of the complete response (Fig. A2b). These results support our interpretation that the temperature-cloud phase relationship is the dominant factor in determining the difference in the two ΔLIG (the difference of the difference) of MIROC4m-LPJ.

**Table A1: A list of AGCM experiments**

| Experiments | Boundary conditions from MIROC4m-LPJ | Temperature dependency of cloud phase | Autoconversion and ice sedimentation |
|---|---|---|---|
| PIvA_AA | PIvA | A | A |
| PIvC_CC | PIvC | C | C |
| LIGvA_AA | LIGvA | A | A |
| LIGvC_CC | LIGvC | C | C |
| PIvA_CA | PIvA | C | A |
| PIvC_AC | PIvC | A | C |
| LIGvA_CA | LIGvA | C | A |
| LIGvC_AC | LIGvC | A | C |
| PIvA_AC | PIvA | A | C |
| PIvC_CA | PIvC | C | A |
| LIGvA_AC | LIGvA | A | C |
| LIGvC_CA | LIGvC | C | A |





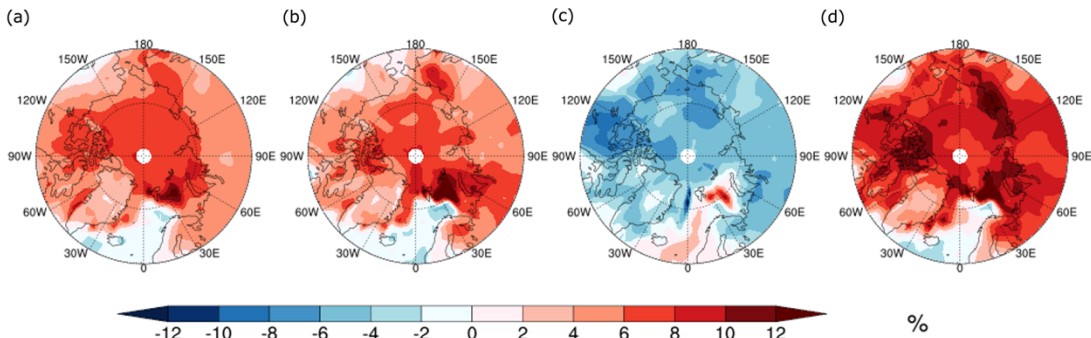

**Figure A1: Difference in low-level cloud amount: (a) PIvC-PIvA by MIROC4m-LPJ; (b) (a) reproduced by the AGCM (PIvC_CC – PIvA_AA); (c) AGCM with only the temperature-phase parameter swapped in (b) (PIvC_AC – PIvA_CA); (d) AGCM with other cloud parameters swapped in (b) (PIvC_CA – PIvA_AC).**

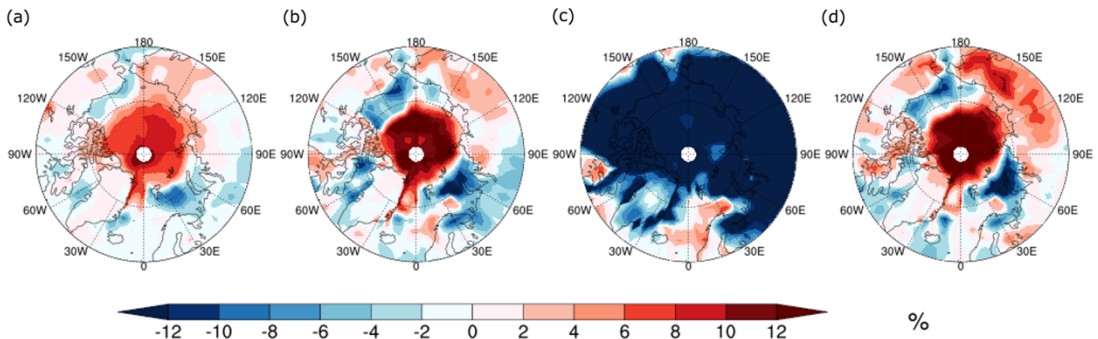

**Figure A2: Difference in low-level cloud amount: (a) ΔLIGvC-ΔLIGvA by MIROC4m-LPJ; (b) (a) reproduced by the AGCM ([LIGvC_CC – LIGvA_AA] – [PIvC_CC – PIvA_AA]); (c) AGCM with only the temperature-phase parameter swapped in (b) ([LIGvC_AC – LIGvA_CA] –[PIvC_AC – PIvA_CA]); (d) AGCM with other cloud parameters swapped in (b) ([LIGvC_CA – LIGvA_AC] – [PIvC_CA – PIvA_AC]).**

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
