# Peer review of "Impact of the temperature-cloud phase relationship on the simulated Arctic warming during the last interglacial"

_EGUsphere, 2025_

## Referee Comment (RC2)

**Impact of the temperature-cloud phase relationship on the simulated Arctic warming during the last interglacial by *Nozomi Arima et al.**

**Summary:**

This manuscript explores how cloud phase representation affects Arctic climate simulations of the Preindustrial and Last Interglacial. The study finds that models allowing more supercooled liquid water simulate stronger warming – particularly in autumn – and reduced sea-ice cover. The manuscript also shows that the inclusion of dynamic vegetation increases warming and reduces LIG sea ice in the Arctic. The paper is mostly well written – it is clear and well structured. I enjoyed reading it. There are, however, several significant issues, some key references are missing, and the use of both present-day and Last Interglacial observations needs improvement.

**Major comments**

**Cloud parameterisations**

Overall, the two cloud parametrisations clearly have substantially different impacts on clouds and temperature in the Arctic over the Preindustrial and Historical periods. This should be dealt with much more clearly prior to their use for the lig127k. Results for Historical simulations should be shown against present-day cloud observations, Arctic temperature, and Arctic sea ice. This does not preclude the main point of this manuscript – that lig127k Arctic temperatures and SIA are sensitive to cloud parameterisation – but it should make the presentation and discussion of the meaning of the results much clearer. Prior to the use of the two parametrisations for the LIG, they should be shown against present-day cloud observations, Arctic temperature, and Arctic sea ice. This could be done in a second Appendix, if the authors prefer not to put this in the main text.

**Fixed-angle calendar**

On the use of a fixed-angle calendar, this paper is largely focused on the correct calculation of energy (and mass) budgets. The re-interpolation from fixed-day to a fixed-angle calendar will lead to incorrect budgets for ice and energy (Sime et al., 2025b). If the authors do include comparison with summer air temperatures in the Arctic, I agree that using fixed-angle JJA averages may be helpful. However, I would like to see a more careful consideration of the impacts of incorrect month lengths on all other averaging – and budgets.

**Observations**

On observations, the use of the Turney and Jones (2010) dataset is not recommended by the PMIP community because of the weak dating constraints. See various papers from Capron et al. If the authors wish to use summer air temperature data for comparison, please use instead the database of CAPE et al. (2006), later updated by Otto-Bliesner for the IPCC, then Guarino et al. (2020), and Sime et al. (2023). The data reference Guarino, M. V., & Sime, L. (2022) is provided below.

**Marine core data**

On marine core data, Vermassen et al. (2023) suggest that the Arctic was likely seasonally seaice-free during the LIG; however, their Arctic age models are currently characterised by significant uncertainty (Razmjooei et al., 2023). For similar reasons, all but one of the Kageyama et al. (2021) Arctic marine core datapoints are marked as being unreliable with respect to dating. Please address this as suggested below.

**Section and Line Comments:**

**L31-42** The Introduction should be modified to better reflect our current understanding of Arctic sea ice and summer temperatures. Please include that Guarino et al. (2020) pointed out that the Arctic during the Last Interglacial was likely sea-ice-free. (Prior to this, Malmierca Vallet et al. (2018) also noted that the Greenland Last Interglacial ice-core water isotope values were most easily explained by a sea-ice-free summer in the Arctic.) Sime et al. (2023) updated this work, estimating the Arctic-wide summer surface air temperature warming at 127 ka to be  $3.7 \pm 1.5$  K, and that the LIG climatological minimum SIA was most likely 1.3 to  $1.5 \times 10^6$  km², which is rather close to the definition of a practically summer ice-free Arctic (a maximum sea-ice extent of less than  $1 \times 10^6$  km²). Read and reference Sime et al. (2025b) *A sea ice free Arctic: Assessment Fast Track abrupt-127k experimental protocol and motivation*.

**L42** This is not correct. See Sime et al. (2023) and Sime et al. (2025b). Six models are occasionally practically sea-ice-free in their first 100 years of lig127k forcing.

**L44-47** Better to read and reference also Sime et al. (2025b), and make it clearer that albedo feedbacks are crucial, and that the inclusion of advanced ice physics such as explicit melt ponds can lead to a better representation of sea ice in models including CESM2 and HadGEM3 (Guarino et al., 2020; Diamond et al., 2024).

**L49-70** I like this section, but find myself also wanting to know what the usual cloud parameterisation schemes used in CMIP6/7 models are. A little background to say which form of parameterisation in cloud scheme A and C are relevant to the CMIP Fast Track community would be very helpful.

Additionally, can you make the relationship between this manuscript and the energy budget findings in Kageyama et al. (2021) clearer? And also relate it to the energy-budget aims of *A sea ice free Arctic: Assessment Fast Track abrupt-127k*.

**L115** The meaning of the comment on which parameterisation is more accurate is very unclear. The two parametrisations clearly have substantially different impacts on clouds and temperature in the Arctic over the Preindustrial and Historical periods. This should be dealt with much more clearly. Prior to the use of the two parametrisations for the LIG, they should be shown against present-day cloud observations, Arctic temperature, and Arctic sea ice.

L175-182 Because the Earth moves fastest near perihelion (when Earth is closest to the sun) and slowest near aphelion (when Earth is farthest from the sun), the use of a fixed-angle calendar causes problems in the calculation of energy budgets (e.g. Otto-Bliesner et al., 2017; Bartlein and Shafer, 2019). This is because the use of a fixed-angle calendar results in months and seasons of unequal day lengths. Given that this paper is largely focused on the correct calculation of energy (and mass) budgets, the re-interpolation from fixed-day to a fixed-angle calendar will lead to incorrect budgets for ice and energy (Sime et al., 2025b). The authors are correct that switching to a fixed-angle calendar can be helpful for comparing seasonal observations across different time periods. However, in the analysis that follows this is not done (see also below). I can see that there are 'monthly' comparisons used for the lig127k output. I'd like to see a more careful consideration of the impacts of incorrect month lengths in this 'monthly' averaging (that it is not possible to compare months in a truly meaningful way between the PI and lig127k), and ideally instead the use of fixed-length output averaging –

perhaps centred on their season/date of interest. For example, using the 15 days either side of

the solstice or equinox, rather than 'September' or 'March' averages. This prevents the temporal stretching and compression which introduces artefacts into energy budgets that are otherwise associated with re-interpolating to a fixed-angle calendar. It would also be helpful for the authors to confirm that they produce any annual averages without angle-based interpolation – i.e. that these are standard (correct) CMIP fixed-length type averages.

**L212, and below** Such large temperature differences between the A and C simulations imply that there should be a preference for one or the other parameterisation on the basis of its match to present-day observations. For example, one can say whether 2.8 °C warmer in November in the PI/present-day is an improvement or not.

**Figure 6, and other places** If removing the attempt to compare 'months', the month (or 30-day period) with the minimum SIC/SIA is otherwise used in most of the papers below – rather than 'September'.

**5.2** The use of the Turney and Jones (2010) dataset is not recommended by PMIP because of the weak dating constraints used (see various papers from Capron et al.). If the authors wish to use summer air temperature data for comparison, please use instead the database of CAPE et al. (2006), later updated by Otto-Bliesner for the IPCC, then Guarino et al. (2020), and Sime et al. (2023). The data reference Guarino, M. V., & Sime, L. (2022) is provided below.

Marine core data Vermassen et al. (2023) suggest that the Arctic was likely seasonally sea-ice-free during the LIG; however, their Arctic age models are currently characterised by significant uncertainty (Razmjooei et al., 2023). For similar reasons, all but one of the Kageyama et al. (2021) Arctic marine core datapoints are marked as being unreliable with respect to dating. For this reason, it is good practice to either (i) not show the Kageyama et al. (2021) datapoints which cannot be reliably dated to the Last Interglacial, or (ii) mark them as 'date unknown' (e.g. see Figure 3 in Sime et al., 2023, and the figures in Kageyama et al., 2021). To show them as dated to the Last Interglacial is misleading.

Section 5.4, and other places A main feedback on clouds in the Arctic is the release of heat in autumn that is stored in the Arctic mixed layer (upper ocean). Less sea ice in summer → more heat absorbed and stored in the mixed layer → released to the atmosphere in autumn, with slower freeze-up and cloud feedbacks. This process is not currently adequately covered in 5.4 or the rest of the paper. The process and its consequences need consideration.

**~L379 referencing etc** Do refer to the energy budget work done in the various Guarino (2020), Diamond (2021/4), Kageyama (2021), and Sime (2025b) papers here too, and rewrite as necessary.

**L386** and thereafter Strangely phrased here. Perhaps start with the obvious – that the differences between cloud parameterisation A and C have an impact in a particular temperature range. This therefore means that impacts will clearly be different between the Southern Ocean and the Arctic.

**L401** Again, odd claim that you can't tell whether a 3 °C warmer Arctic in November (and other temperature and cloud changes) in the PI/present-day is better or not. Best to rewrite these sections once this is clearer to the authors.

L. Sime

**Data reference:**

Guarino, M. V., & Sime, L. (2022). Last Interglacial summer air temperature observations for the Arctic (Version 1.0) [Data set]. NERC EDS UK Polar Data

Centre. https://doi.org/10.5285/9AB58D27-596A-472C-A13E-2DCD68612082

**References:**

Sime, Louise C., Diamond, Rachel, Stepanek, Christian, Brierley, Chris, Schroeder, David, Kageyama, Masa, Malmierca-Vallet, Irene, Blockley, Ed, West, Alex, Feltham, Danny, Ridley, Jeff, Braconnot, Pascale, Williams, Charles J. R., Shi, Xiaoxu, Otto-Bliesner, Bette L., Macarewich, Sophia I., Ramos Buarque, Silvana, Zhang, Qiong, LeGrande, Allegra, Zheng, Weipeng, Jiang, Dabang, Morozova, Polina, Guo, Chuncheng, Zhang, Zhongshi, Yeung, Nicholas, Menviel, Laurie, Narayanasetti, Sandeep, Reeves, Olivia, Pollock, Matthew, Zhao, Anni. (2025b) A sea ice free Arctic: Assessment Fast Track abrupt-127k experimental protocol and motivation [in review]. *EGUsphere* [preprint], (). pp. doi:10.5194/egusphere-2025-3531

Sime, Louise C., Sivankutty, Rahul, Malmierca-Vallet, Irene, Goursaud Oger, Sentia, LeGrande, Allegra N., McClymont, Erin L., de Boer, Agatha, Cauquoin, Alexandre, Werner, Martin. (2025a) More modest peak temperatures during the Last Interglacial for both Greenland and Antarctica suggested by multi-model isotope simulations [preprint]. Climate of the Past [in review], (). pp. 10.5194/egusphere-2025-288

Diamond, Rachel, Schroeder, David, Sime, Louise C., Ridley, Jeff, Feltham, Danny. (2024) The significance of the melt-pond scheme in a CMIP6 global climate model. *Journal of Climate*, 37 (). pp. 10.1175/JCLI-D-22-0902.1

Sime, Louise C., Sivankutty, Rahul, Malmierca Vallet, Irene, de Boer, Agatha M., Sicard, Marie. (2023) Summer surface air temperature proxies point to near-sea-ice-free conditions in the Arctic at 127 ka. Climate of the Past, 19 (). pp. 10.5194/cp-19-883-2023

Guarino, Maria Vittoria, Sime, Louise C., Schröeder, David, Malmierca Vallet, Irene, Rosenblum, Erica, Ringer, Mark, Ridley, Jeff, Feltham, Danny, Bitz, Cecilia, Steig, Eric J., Wolff, Eric, Stroeve, Julienne, Sellar, Alistair. (2020) Sea-ice-free Arctic during the Last Interglacial supports fast future loss. *Nature Climate Change*, 10 (). pp. 10.1038/s41558-020-0865-2

Malmierca Vallet, Irene, Sime, Louise C., Tindall, Julia C, Capron, Emilie, Valdes, Paul J, Vinther, Bo M. (2018) Simulating the Last Interglacial Greenland stable water isotope peak: the role of Arctic sea ice changes. *Quaternary Science Reviews*, 198 (). pp. 10.1016/j.quascirev.2018.07.027

Capron, Emilie, Govin, Aline, Stone, Emma J., Masson-Delmotte, Valérie, Mulitza, Stefan, Otto-Bliesner, Bette, Sime, Louise C., Waelbroeck, Claire, Wolff, Eric W.. (2014) Temporal and spatial structure of multi-millennial temperature changes at high latitudes during the Last Interglacial. Quaternary Science Reviews, 103 (). pp. 10.1016/j.quascirev.2014.08.018

Kageyama, Masa, Sime, Louise C., Sicard, Marie, Guarino, Maria Vittoria, de Vernal, Anne, Schroeder, David, Stein, Ruediger, Malmierca Vallet, Irene, Abe-Ouchi, Ayako, Bitz, Cecilia, Braconnot, Pascale, Brady, Esther, Chamberlain, Matthew A., Feltham, Danny, Guo, Chuncheng, Lohmann, Gerrit, Meissner, Katrin, Menviel, Laurie, Morozova, Polina, Nisancioglu, Kerim H., Otto-Bliesner, Bette, O'ishi, Ryouta, Sherriff-Tadano, Sam, Stroeve, Julienne,

Shi, Xiaoxu, Sun, Bo, Volodin, Evgeny, Yeung, Nicholas, Zhang, Qiong, Zhang, Zhongshi, Ziehn, Tilo. (2021) A multi-model CMIP6-PMIP4 study of Arctic sea ice at 127ka: Sea ice data compilation and model differences. Climate of the Past, 17 (). 26 pp. 10.5194/cp-17-37-2021

Razmjooei, M. J., Henderiks, J., Coxall, H. K., Baumann, K.-H., Vermassen, F., Jakobsson, M., Niessen, F., and O'Regan, M.: Revision of the Quaternary calcareous nannofossil biochronology of Arctic Ocean sediments, Quaternary Science Reviews, 321, 108 3846

---

## Referee Comment (RC4)

**Review of the manuscript "Impact of the temperature-cloud phase relationship on the simulatedArctic warming during the last interglacial" by Arima and Yoshimori et al.**

**Overview**

In their study Arima and Yoshimori et al. address the problem that many climate models are too insensitive to the radiative forcing of the Last Interglacial (LIG) to reproduce current inferences on the state of the Arctic Cryosphere. The literature has addressed many hypotheses and studied how retuning parameterizations of climate models optimized for current climate could reduce some of the biases of models with regard to inferences from the geologic record. In their study Arima and Yoshimori et al. shortly refer to previous work that aims at explaining potential causes for limited model sensitivity when simulationg LIG climate (including vegetation dynamics and melt-pond scheme). Based on this motivation the authors focus on another aspect of climate model formulation and demonstrate, how changes in cloud parameter sets, in combination with and without dynamic vegetation, can lead to differences in simulated climates. The authors find that increasing the availability of supercooled liquid water warms the Arctic and reduces LIG summer sea ice via different behavior of low-level clouds.

The research presented by Arima and Yoshimori et al. presents a prime example of study that a) employs paleoclimate as a laboratory to understand potential reasons for model biases with respect to our inference from the geologic record, b) applies that understanding towards developing a modified climate model, and c) critically evaluates the changes in the context of other uncertainties in the model. I find the presented research both important and generally very well written. I recommend it for publication in Climate of the Past after addressing or rebutting a number of comments (mostly minor) that I suggest for consideration by the authors when creating a revised manuscript. Please refer to details below.

**Overarching comments**

The relevant mechanism that is the focus of this study is explained in detail in the introduction and mostly supported by referenced relevant literature, contributing to the comprehension of the presented research. Exceptions to this rule are, in my humble opinion, the explanation of the technical details in Section 2. From statements on pages 4 and 5 I interpret that the actual relevant model parameter to change is T\_ice, and that the other two parameters are merely modified to retune the simulated climate to maintain sufficient model skill for some observed modern climate patterns. Presenting this information further up in Section 2.2 may simplify comprehension on the side of your readership. Highlighting T\_ice as the relevant change (maybe even in Table 1) would make this even clearer. Lots of parameters are referenced (" $\alpha$ ,  $\beta$ ,  $\gamma$  and C\_c" are constants"; "V\_0 and  $\delta$  are constants"). Please make clear in the text whether these are the same as those that you specifically target in your simulation, and also describe what the other symbols refer to (unless they are not of interest to this study, which would then pose the question why they are mentioned here).

Some formulations and text sections remain unclear to me and are (in my personal opinion) overly complicated. Examples:

line 103: "We note that the parameter set "B" referred to the parameter set A applied to a different model version in their study as well as in Sherriff-Tadano and Abe-Ouchi (2020), and we retain the names "A" and "C" in this study." Here a reader, who does not already know details of the model parameter sets, may feel overwhelmed. Given that the research presented here should be relevant well beyond the MIROC community I suggest to aim for easier comprehension. The whole paragraph seems to be providing very detailed technical information without providing an explanation at a higher flight level.

The authors provide a clear description of the expected impact of changes in selected parameter values (line 140f). I think that highlighting this aspect more, maybe even further up, may simplify reading the various technical details (that are important nonetheless). Ideally, one could add in the discussion section later on whether there were any surprises found in the modelled results; i.e., where there any simulated changes in climate characteristics that differed from what the authors initially expected?

It may be a personal limitation on my side, but I was sometimes a bit confused on the parameter set names "A" and "C". The relationship of the other parameter set, "B", to "A", and what its difference and relevance to the work at hand might be, remains unclear. If there is no good reason to keep the names as they are (maybe "A" and "C" carry a specific meaning in the MIROC community that must be maintained for clarity) then I would maybe come up with clearer names that aim for simplicity and comprehension within this current manuscript.

A bit more explanation regarding the model-data comparison and the analysis of contributions of different actors in the energy balance to the overall temperature change would be helpful. While at first sight it appears that the energy balance analysis may be done based on results from the fully coupled atmosphere-ocean simulation including sea surface, some statements in the text (subsurface heat from the soil rather than from the ocean) and the selection of the proxy data base may hint that the focus is on atmosphere model output (standalone atmosphere model, see lines 169ff) alone. If so, what is the motivation? Please clarify the text accordingly.

Terminology is not consistent across the manuscript, in particular Table 5 and legends of various figures could be improved in this regard. Very important, please make sure that each and every parameter on pages 7 and 8 is clearly defined and explained, ideally also featuring the respective physical symbol in Table 5. This is important in particular for alpha, since that parameter has different meanings between pages 6/7 (albedo) and 4/5 (autoconversion rate).

When referring to the use of the model in previous studies, I suggest to provide a bit of detail in how far the referenced previous work is specifically relevant to the current work. If you did so, then the relevant statements would be a bit more connected to your work. For example: line 99 "This vegetation-coupled climate model has also been used in previous studies (e.g., Hirose et al., 2025; O'ishi et al., 2021)." and line 91 "This climate model runs computationally very efficiently and has been used in many previous studies (e.g., Chan and Abe-Ouchi, 2020; Kuniyoshi et al., 2022; Sherriff-Tadano et al., 2023)." A bit more details may be informative towards judging the specific value of this model for the research at hand.

For some statements that are based on referenced literature more appropriate references could be chosen. For example, statements on the state of the LIG could be made based on primary literature from multi-model analysis in PMIP or proxy-based inferences, many of which may be cited by Gulev et al. (2021), rather than refering to Gulev et al. (2021) alone (choice is of course subjective and this is merely a suggestion).

For the statement "given that the Arctic is currently warming much faster than the rest of the world, and sea ice is decreasing dramatically." one could provide a reference to observations or reanalyses (again, subjective point of view). Not sure whether the statement "Arctic amplification reaches its peak" (line 64) is covered by the reference mentioned in the same line before. If applicable, one could add more specific references here.

**Specific comments:**

line 41: add a comma after "assemblages"? (not sure)

line 69: capitalize Last Glacial Maximum?

line 72: Please address in your dicussion to which extend keeping the number of ice-nucleating particles fixed could impact the results.

line 73: the meaning of the following statement remains unclear to me, please check: "Additionally, we would like to discuss the magnitude of this effect against a range of model spreads."

line 96: grid -> grid cell?

line 155: "the most dominant vegetation type simulated by PlvA over the last 100 years for each grid is prescribed in both PlfA and LlGfA experiments" - was this done on grid cell level or at global scale? Does dominant refer to what is shown in Figure 3? This is not clear to me.

line 185: "whose total sum amounts to the simulated surface temperature change under an excellent approximation" - Do you maybe mean here: "whose total sum is an excellent approximation to the simulated surface temperature change?"

line 190f: "and Q is the heat storage rate in the subsurface (e.g., heat conduction into the soil layers)". Does this analysis only refer to the standalone atmosphere simulation that you mention a bit further up? If not, then heat storage in the ocean may be an at least equally relevant process to mention here?

line 198: type setting of parameters evap, sens and subsurf could be improved (use one letter symbols instead)?

line 280: "It is important to note that the LIG temperature reconstruction by Turney and Jones (2010) is compiled from the locally warmest time within a wide period of approximately 13,000 years, from 129 to 116 kaBP. Thus, it tends to overestimate the 127 kaBP warming rather than underestimate it". In the text one nearly overreads that you actually also make a data comparison with Capron et al. (2017). You mention this only in the caption to Fig. 8. I suggest to make the use of two different proxy data sets clearer at this location. Furthermore, you could elaborate on arguments why the comparison to Capron et I. (2017) may be more robust, and you could reflect on differences in the model-data comparison that may arise from using any of those two data sources. Capron et al. (2017) provide some arguments why Turney and Jones (2010) should (maybe) not be used for an evaluation of 127k. If the authors see good reasons for comparing against Turney and Jones (2010) (while they are aware of some of the drawbacks as explicitly stated here), then it would be good to provide at a suitable location of the manuscript a motivation / justification for that decision.

line 286: add a dot after the bracket; grids-> grid cells?

line 292: delete dot after "Figures."

line 304: not sure whether the term "grossly" is the best choice here, it has a very negative conotation.

line 314: When reading this I already had forgotten the definition of the abbrevation CRE (line 226). This term does not appear too often, one could consider to write it out for clarity.

line 328: capitalize archipelago? (also line 404)

line 329ff: While the argumentation seems plausible, I am asking myself whether the simulation and model output would allow a further test of this statement. Maybe it would be already illustrative to extend Figure 11 by showing more preceeding and subsequent months to illustrate when and where the relationship breaks?

line 344: delete "The"? Use an em-dash (or similar) for the "range"-sign in October - December?

line 348: keep minus sign directly in front of the value to avoid confusion

line 370: "We added four values of summer (July-September) sea ice area (LIGvA, LIGvC, LIGfA, and LIGfC) to the multi-model data shown in Fig. 13." I suggest to refer here to the original publication for clarity.

line 442: "for the future simulations" - do you refer here to simulations of future climate or to future (i.e. work in progress) simulations of LIG climate?

line 457: "importing the model to" -> "deploying the model at"?

Figure 1: caption could be more informative. In particular, a description of the meaning of the vertical gray dashed bars (relevant temperature ranges for coexistence of solid and fluid cloud droplets in parameter sets A and C), and a reference to relevant mathematical symbols (e.g. T\_ice), would make information in the figure more self-contained.

Figure 3: Are these the dominant vegetation types the text speaks of? I.e. are there more vegetation types per grid cell, but you show here only the dominant one? Please clarify the text accordingly.

Figure 4: "Please refer to Table 5 for a description of each component" - I note that keys in the legend differ from those used in Table 5. This could be harmonized.

Figure 8: I am at a loss what is the difference between this figure and Fig. S4. The modelled temperature anomalies shown for these two figures are obviously different, but figure captions appear identical to me (except for a typo). Could it be that the caption in S4 is wrong and should refer to LIGfA (=LIGfA-PIfA); (b) LIGfC (=LIGfC-PIfC) (replace any "v" with an "f").

Figure 9: increasing font size may lead to easier readability.

Figure 10: Subfigure b seems to have a partial frame, I guess this is an artifact.

Figure 11: increase font size

Figure 12: increase font size

Figure 13: You use different abbreviations for Northern Hemisphere in x-axis label and caption

**Appendix:**

Is there a good reason to have a rather short appendix rather than implementing this information directly in the main text?

Figure A1: Increase font size

Figure A2: Increase font size

**Supplement:**

For those Figures that are similar to Figures in the main text it could be made clearer in the captions what is the difference to the Figure in the main text. In one case absence of such clear statements, in combination with a series of typos ("v" instead of "f") left me clueless regarding the data shown.

Explain the abbreviations (PIvA, PIvC, LIGvA, LIGvC) used in Fig. S1 within the supplement. While you refer to the main text for details, the location where you reference Fig. S1 first (Section 2, line 139) does not directly provide these definitions, which rather follow later in Section 3, thereby leading to unclarity of their meaning at this place.

Not sure whether there is a more meaningful location for the data availability disclaimer (currently in line 25ff). Please check whether it must be put here rather than in the main manuscript's corresponding data availability and disclaimer sections.

Fig. S4: potentially an error in the caption, see my remark at Fig. 8

line 15: remove brackets around the unit? grids -> grid cells?

line 22: not sure whether "crude" is the right term here (it is also used at some other locations at the manuscript, please also check there)